# Functional Versatility of the Human 2-Oxoadipate Dehydrogenase in the L-Lysine Degradation Pathway toward Its Non-Cognate Substrate 2-Oxopimelic Acid

**DOI:** 10.3390/ijms23158213

**Published:** 2022-07-26

**Authors:** Natalia S. Nemeria, Balint Nagy, Roberto Sanchez, Xu Zhang, João Leandro, Attila Ambrus, Sander M. Houten, Frank Jordan

**Affiliations:** 1Department of Chemistry, Rutgers University-Newark, Newark, NJ 07102-1811, USA; xz364@scarletmail.rutgers.edu; 2Department of Biochemistry, Institute of Biochemistry and Molecular Biology, Semmelweis University, 1082 Budapest, Hungary; nagy.balint@med.semmelweis-univ.hu (B.N.); ambrus.attila@med.semmelweis-univ.hu (A.A.); 3Icahn Institute for Data Science and Genomic Technology, Icahn School of Medicine at Mount Sinai, New York, NY 10029-6501, USA; roberto.sanchez@mssm.edu (R.S.); joao.leandro@mssm.edu (J.L.); sander.houten@mssm.edu (S.M.H.)

**Keywords:** 2-oxoadipate dehydrogenase, L-lysine degradation pathway, 2-oxopimelate substrate, 2-oxoadipate dehydrogenase complex, E1a promiscuity, H_2_O_2_ production, rare E1a variants

## Abstract

The human 2-oxoadipate dehydrogenase complex (OADHc) in L-lysine catabolism is involved in the oxidative decarboxylation of 2-oxoadipate (OA) to glutaryl-CoA and NADH (+H^+^). Genetic findings have linked the *DHTKD1* encoding 2-oxoadipate dehydrogenase (E1a), the first component of the OADHc, to pathogenesis of AMOXAD, eosinophilic esophagitis (EoE), and several neurodegenerative diseases. A multipronged approach, including circular dichroism spectroscopy, Fourier Transform Mass Spectrometry, and computational approaches, was applied to provide novel insight into the mechanism and functional versatility of the OADHc. The results demonstrate that E1a oxidizes a non-cognate substrate 2-oxopimelate (OP) as well as OA through the decarboxylation step, but the OADHc was 100-times less effective in reactions producing adipoyl-CoA and NADH from the dihydrolipoamide succinyltransferase (E2o) and dihydrolipoamide dehydrogenase (E3). The results revealed that the E2o is capable of producing succinyl-CoA, glutaryl-CoA, and adipoyl-CoA. The important conclusions are the identification of: (i) the functional promiscuity of E1a and (ii) the ability of the E2o to form acyl-CoA products derived from homologous 2-oxo acids with five, six, and even seven carbon atoms. The findings add to our understanding of both the OADHc function in the L-lysine degradative pathway and of the molecular mechanisms leading to the pathogenesis associated with DHTKD1 variants.

## 1. Introduction

The human 2-oxoadipate dehydrogenase (E1a, also known as DHTKD1) is a mitochondrial protein in the L-lysine degradative pathway which is involved in the formation of glutaryl-CoA and NADH (+H^+^) from 2-oxoadipate (OA) [1,2,3,4] and is critical for mitochondrial metabolism [5,6,7]. It was recognized that E1a could assemble into a 2-oxoadipate dehydrogenase multienzyme complex (OADHc) by sharing the dihydrolipoamide succinyltransferase (E2o) and dihydrolipoamide dehydrogenase (E3) components with 2-oxoglutarate dehydrogenase (E1o) of the 2-oxoglutarate dehydrogenase complex (OGDHc), suggesting a crosstalk between the OGDHc in the tricarboxylic acid (TCA) cycle and OADHc in L-lysine catabolism [2,3,4]. Genetic studies have linked mutations in the *DHTKD1* gene encoding E1a to the (neuro) pathogenesis of several disorders: alpha-aminoadipic and alpha-ketoadipic aciduria (AMOXAD), an inborn error of metabolism of unknown clinical significance [8,9,10]; Charcot-Marie-Tooth disease type 2Q (CMT2Q), a disease of the peripheral nervous system [11,12]; eosinophilic esophagitis (EoE), a chronic allergic disorder [6], and infantile-onset spinal muscular atrophy (SMA) [13]. Rare heterozygous *DHTKD1 encoded* DHTKD1 variants were recently identified in patients with amyotrophic lateral sclerosis (ALS), a progressive neurodegenerative disorder [14]. Pharmacological inhibition of E1a has been proposed as a strategy for substrate reduction therapy to treat glutaric aciduria type 1 (GA1) [15]. However, there are currently no studies in vitro for most of the identified DHTKD1 variants. Instead, evidence in silico and in vivo is employed as support for their pathogenicity. Genetic findings suggest that an in-depth understanding of the E1a function is needed.

Here we have applied a multipronged approach, including kinetic analysis, fluorescence and circular dichroism spectroscopy, Fourier Transform Mass Spectrometry, and computational approaches to provide novel insight into the mechanism and functional versatility of the OADHc. Studies provide compelling evidence indicating that the human OADHc could use the non-natural alternative substrate 2-oxopimelic acid (OP, also known as α-ketopimelic acid or 2-oxoheptanedioic acid), a metabolite currently unknown in human metabolism [16], enabling us to suggest a mechanism for OP transformation by OADHc in vitro. Currently, the functional and metabolic roles of OP and pimelic acid (PA, also known as heptanedioic acid) are better understood in prokaryotes, fungi, and higher plants. The OP has been reported as a potent inhibitor of the *E. coli* [17] and *Mycobacterium tuberculosis* [18] dihydrodipicolinate synthase, a unique enzyme in the diaminopimelate pathway of lysine biosynthesis, which is essential for bacterial growth and survival. The thioesters of PA, pimeloyl-CoA, and pimeloyl-acyl carrier protein (pimeloyl-ACP) are known intermediates of the biotin synthetic pathways in bacteria, archaea, fungi, and plants [19,20,21,22], where distinct genes and enzymes have been shown to be involved in the synthesis of the pimeloyl moiety. Free PA was identified as an intermediate of the biotin synthetic pathway in the gram-positive bacterium *Bacillus subtilis* [19]. In humans, PA is found in serum and urine, suggesting that it is most likely derived from β-oxidation of odd-chain linear saturated dicarboxylic fatty acids such as pentadecanedioic acid (C15) and heptadecanedioic acid (C17), which are generated by ω-oxidation of the mono-carboxylic odd-chain fatty acids [23,24]. Moreover, in humans, the mitochondrial 2-oxodicarboxylate carrier (ODC) has been identified, which catalyzes the transport of the C5-C7 2-oxodicarboxylates such as 2-oxoadipate (OA, C6), 2-oxoglutarate (OG, C5) and to a lesser extent, PA (C7) and OP (C7) across the inner membranes of mitochondria [25]. In the search for human E1a inhibitors to treat GA I, it was shown that E1a by itself displayed activity with OP, similar to that with natural substrate OA, while the phosphonic acid analogue of OP, adipoyl-phosphonic acid, was shown to behave as a mechanism-based, tight-binding inhibitor of E1a and of OADHc in vitro and of E1a in HEK293 cells [16]. However, the source of OP in humans remains elusive. Here we identified a striking case of promiscuity in human E1a, which oxidized 2-oxoglutarate (OG, C5), 2-oxoadipate (OA, C6) and 2-oxopimelic acid (OP, C7) with the formation of succinyl-CoA, glutaryl-CoA and adipoyl-CoA, respectively. However, a distinct preference for longer carbon chain (C6 and C7) substrates was found, thus providing a broader and more in-depth understanding of molecular pathways associated with E1a and OADHc. We also demonstrated a strong correlation between the OA and OP oxidation by E1a variants with mutations associated with AMOXAD and EoE, thus providing a better understanding of their pathogenic mechanisms.

## 2. Results and Discussion

### 2.1. The E1a Oxidizes 2-Oxopimelic Acid at Rates Similar to the Rates with Its Substrate 2-Oxoadipate through Decarboxylation and Enamine Intermediate Formation

It was reported previously that OADHc oxidizes 2-oxoglutarate (OG, C5) in addition to its natural substrate 2-oxoadipate (C6) in vitro, however, with a 49-fold preference for OA over OG [2]. The high specificity of the E1a toward OA was demonstrated in a HEK-293 cell lysate model system where the effect of transient overexpression of the E1o, E1a, E2o and E3 components was determined on the activity of OGDHc and OADHc [4]. It was demonstrated that in HEK-293 cell lysates the oxidative decarboxylation of OA is ~40 times lower compared to OG and could be increased upon transfection of *DHTKD1* and/or *OGDH* while the activity with OG could be increased upon transfection of *OGDH* solely [4]. Importantly, experiments in vitro and in cell model systems revealed that OGDHc and E1a could be assembled into a hybrid 2-oxo acid dehydrogenase complex containing E1a, E1o, E2o and E3 components from two distinct metabolic pathways [4]. In the search for E1a inhibitors to treat GA I, it was demonstrated that human E1a can decarboxylate the non-natural substrate OP (C7) in addition to OA (C6) and OG (C5), suggesting its promiscuity [16]. However, no detailed kinetic analysis of the OP oxidation by OADHc has been reported so far. The question of why the human E1a could not differentiate between the non-natural substrate OP and its natural substrate OA has been raised. This is a fundamental question to explore the origins of the E1a function and to understand the pathogenicity of the significantly increased amount of the rare DHTKD1 variants associated with the neurodegenerative diseases.

Below, we applied a multipronged approach developed and employed in the Rutgers group to analyze each step in the mechanism of the OADHc with OP as a substrate, assuming that the transformation of OP by the OADHc would be similar to that with the OA pathway. There are several intermediates on the OADHc pathway converting OA to glutaryl-CoA according to the detailed chemistry presented in Equations (1)–(4) and in Figure 1:2-oxoadopate + E1a → C2-(α-hydroxy)-δ-carboxybutylidene-ThDP-E1a(E1a-Thdp-enamine intermediate) + CO_2_(1)
C2-(α-hydroxy)-δ-carboxybutylidene-ThDP-E1a + lipoyl-E2o → *S*8-glutaryldihydrolipoyl-E2o (reductive glutarylation)(2)
*S*8-glutaryldihydrolipoyl-E2o + CoA → glutaryl-CoA + dihydrolipoyl-E2o(3)
Dihydrolipoyl-E2o + E3 + NAD^+^ → lipoly-E2o + NADH^+^ (+H^+^)(4)

To demonstrate whether the OP (C7), a one-carbon longer analogue of the natural substrate OA (C6), is transformed by the OADHc, the E1a-specific activities with OA and OP were re-examined first. It needs to be noted that in the paper reported earlier where OP was identified as an E1a substrate for the first time, the E1a-specific activity with OP was measured in the reaction assay adopted for high-throughput screening of the inhibitors rather than for kinetic analysis with the progress curves recorded for over 30 min [16]. Briefly, in the E1-specific assay, the E2o and E3 components are replaced by an external chromophoric two-electron acceptor, 2,6-dichlorophenolindophenol (DCPIP), whose reduction is monitored at 600 nm, thus informing about substrate binding in the active centers of E1a (*K_m_)* and the formation of the enamine intermediate (*k*_3_ in Figure 1). Similar E1a-specific activities were displayed toward OA (1.86 μmol·min^−1^· mg E1a^−1^) and OP (1.56 μmol· min^−1^·mg E1a^−1^) (Table 1, upper right quadrant). However, the calculated value of *K_m_,_OP_* of 37 μM was ~8-fold higher compared to that of *K_m,OA_* (4.9 μM), both measured in the E1a-specific assay, thus lowering the E1a specific activity (*k_cat_*/*K_m_*) toward OP by ~9-fold compared to OA (Table 1, upper right quadrant). In comparison, a value of *K_m_* = 11.3 mM was reported for OG earlier [2], indicating that OA and OP with a longer carbon chain are better substrates of the E1a compared to OG. This is also evident from a comparison of the kinetic constants (*k_3_* in Figure 1) determined in the E1a-specific assay: 6.5 s^−^^1^ (OA), 5.4 s^−^^1^ (OP), and 0.12 s^−^^1^ (OG). Hence, the E1a could accept OP with a one-carbon longer chain compared to its natural substrate OA with no significant effect on the rate of decarboxylation, leading to the formation of the E1a-ThDP-enamine intermediate.

### 2.2. The E1a-ThDP-Enamine Intermediate Formed on Decarboxylation of OP Is Catalytically Competent in the E1a-Side Reactions and Could Be an Efficient Source of H_2_O_2_ Production

The E1a-ThDP-enamine intermediate formed on decarboxylation of OA or OP has three alternative pathways, two nonoxidative and the third one oxidative (Figure 1). In the first nonoxidative reaction, the E1a behaves as 2-oxopimelate decarboxylase, yielding adipic acid semialdehyde from the nonoxidative decarboxylation of OP. The rate of adipic acid semialdehyde formation by E1a of 5.7 nmol min^−1^·mg E1a^−1^ was not different from that reported for glutaric acid semialdehyde formation from OA (6.05 nmol·min^−1^·mg E1a^−1^) [26] and from succinic acid semialdehyde formation from OG (5.31 ± 0.01 nmol min^−1^·mg E1a^−1^, *k_cat_* = 0.018 ± 0.003 s^−1^), revealing a low E1a activity in this reaction with no obvious effect on elongation of the substrate length from OG to OP (See Table 1, bottom lines for adipic acid and glutaric acid semialdehyde formation; see Figure 1, upper right quadrant for comparison of rate constants.

The second side-reaction is a non-physiological carboligation reaction (C-C bond formation), where the E1a-ThDP-enamine intermediate formed on decarboxylation of OP adds to an aldehyde acceptor such as glyoxylate to form (*S*)-2-hydroxy-3-oxo-octanedioic acid, a chiral α-ketol carboligation product (Table 1, lower left quadrant; Figure 1, lower right quadrant). The activities of 2-oxoadipate: glyoxylate and 2-oxopimelate:glyoxylate reactions determined by circular dichroism (measured at CD_280_, a signature for chiral α-ketols) of 1.04 μmol·min^−1^·mg E1a^−1^ (OA) and of 1.46 μmol·min^−1^·mg E1a^−1^ (OP) were similar as well as the calculated rate constants, however, the catalytic efficiency (*k_cat_*/*K_m_*) of 138 × 10^3^ M^−1^ s^−1^ (OP) was ~5.3-fold lower compared to that with OA (736 × 10^3^ M^−1^ s^−1^) due to a difference in values of *K_m_* (see Table 1 Legend for values of *K_m,OA_ and K_m,OP_).* For comparison, the carboligation activity of 0.184 ± 0.020 μmol·min^−1^·mg E1a^−1^ was reported for E1a with OG as substrate (*k_cat_* = 0.64 s^−1^; *k_cat_/K_mOG_* = 0.057 × 10^3^ M^−1^ s^−1^) [2], again demonstrating that OG is a poor substrate for E1a. Next, ethyl glyoxylate and methyl-glyoxal were also tested as possible acceptor substrates. While no chiral α-ketol carboligation product was detected by CD with methylglyoxal as acceptor, formation of (*S*)-8-ethoxy-7-hydroxy-6,8-dioxo-octanoic acid by E1a from OP and ethyl glyoxylate as acceptor was indeed in evidence (Figure 1). These findings suggest that the E1a-catalyzed formation of α-ketol carboligation products could create a novel approach for the biocatalytic synthesis of chiral α-hydroxy ketones, valuable intermediates in the pharmaceutical industry (see SI Results and Discussion; SI Scheme S1). The third side-reaction derived from the E1a-enamine intermediate is its oxidation by molecular O_2_ and the concomitant production of the reactive oxygen species, superoxide and H_2_O_2_ (Figure 1). The superoxide anion radical generated could be detected through an assay of H_2_O_2_ formation, a product of dismutation of superoxide radical anion either spontaneously or by the superoxide dismutase present in the reaction assay. Again, similar activities and reaction rates of H_2_O_2_ production by E1a were determined with OA and OP (Table 1, lower right quadrant), however, the catalytic efficiency *k_cat_*/*K_m_*, of 0.78 × 10^3^ M^−1^ s^−1^ for OP was ~7.6-fold lower compared to that with OA (*k_cat_*/*K_m_* = 5.9 × 10^3^ M^−1^ s^−1^) due to its weaker binding. To assess the contribution of E1a to superoxide and H_2_O_2_ production from OA, OP and OG, we next compared the H_2_O_2_-generating activity for the three substrates. Earlier, we reported an H_2_O_2_ producing activity of 0.11 nmol·min^−1^·mg E1a^−1^ (*k_cat_/K_m_* = 0.0035 × 10^3^ M^−1^ s^−1^) for E1a with OG [2]. The data indicated that E1a is at least 223-times more efficient in H_2_O_2_ production from OP and 1686-times more efficient in H_2_O_2_ production from OA compared to that from OG. Next, on comparison with the catalytic efficiency of the human E1o toward H_2_O_2_ production from OG (0.062 × 10^3^ M^−1^ s^−1^) and from OA (0.027 × 10^3^ M^−1^ s^−1^) [3], it has become evident that E1a is a better source of H_2_O_2_ production from both OA and OP compared to E1o with its natural substrate OG.

### 2.3. Generation of NADH from OP Is Strongly Diminished Due to the Diminished Rate of Reductive Transfer of the Adipoyl Moiety from the E1a to the E2o Component Affected by One-Carbon Substrate Elongation

Next, the success of the assembly of E1a with the E2o and E3 components into OADHc, which would result in NADH production from OP, was assessed. The specific activity of OADHc with an OP of 0.012 μmol·min^−1^·mg E1a^−1^ was ~115-fold lower than that with OA (1.38 μmol·min^−1^·mg E1a^−1^) (Table 1, upper left quadrant). This suggests that transfer of the adipoyl group between the E1a and E2o components and/or the formation of the adipoyl-CoA product in the active centers of E2o could be affected. The rate of reductive adipoylation of E2o by E1a and OP was quantified by using a model reaction where E2o was replaced by its lipoyl domain (LDo) with lipoic acid covalently attached to the LDo by the lipoate protein ligase in vitro (Figure 2A) [27]. On decarboxylation of OP by E1a in the presence of the LDo, both un-adipoylated (mass of 11,246.2 Da) and adipoylated (mass of 11,376.2 Da) forms of the LDo could be detected by mass spectrometry (MS) at different times of reaction, enabling the calculation of *k_adipoyl_* = 0.034 s^−1^ (Figure 2B, *k_5_* in Figure 1). The rate constant for this reaction was comparable to the *k_cat_* of 0.04 s^−1^ for NADH formation by OADHc from OP, suggesting that adipoyl transfer between the E1a and E2o could be affected by one-carbon substrate elongation from OA to OP. For comparison, the values of *k_succinyl_* of 6.3 s^−1^ for succinyl transfer and *k_glutaryl_* of 7.6 s^−1^ for glutaryl transfer between E1a and E2o were about 185-fold and 224-fold higher, respectively, compared to adipoyl transfer under similar conditions (Figure 2B,E). The data present evidence that the E2o component of the OGDH in the TCA cycle can produce acyl-CoA thioesters of different lengths, such as succinyl-CoA (C4), glutaryl-CoA (C5) and, less efficiently, adipoyl-CoA (C6), indicating its promiscuity while providing efficient discrimination versus substrates. The next step in the mechanism subsequent to reductive adipoylation of the lip-E2o is the transfer of the adipoyl group from the *S*8-adipoyldihydrolipoyl-E2o to CoA in the active centers of E2o. The reaction rate of this step could be quantified via FT-MS detection of the adipoyl-CoA formed in the reaction of OADHc with OP (*k*_6_ in Figure 1). After 60 min of reaction with OP, almost 90% of the CoA present in the reaction mixture was converted to adipoyl-CoA by OADHc, thereby confirming the success of the overall reaction (Figure 3, Table 1). The calculated second-order rate constant of 0.003 μM^−1^ s^−1^ was ~83-fold slower compared to the second-order rate constant of 0.245 μM^−1^ s^−1^ reported for OA (*k*_6_ in Figure 1) [26], presenting evidence that transfer of the adipoyl group to CoA in the active centers of E2o is affected by the one-carbon substrate elongation in OP. These findings confirmed those reported by us earlier [26] that the rate-limiting step is determined by the E2o component at two steps in the mechanism: by the transfer of the acyl group between the E1a and E2o components, as was generally accepted, and by the transfer of the acyl group from the acyldihydrolipoyl-E2o to CoA in the active centers of E2o. At the same time, recycling of dihydrolipoyl-E2o to lipoyl-E2o proceeded faster, i.e., was not part of the rate limitation.

### 2.4. Binding Models of E1a with OP and of E2o with S8-Adipoyldihydrolipoamide in the Active Centers

In addition to the biochemical findings, induced fit docking was carried out to fit the OP into the active center of human E1a. No significant differences were found between the OP and OA binding in the active center of E1a (Figure 4, top). Similar amino acids were identified poised to interact with OA and OP, showing that one carboxyl group of the substrate forms hydrogen bonds with His435 and His708, while the other carboxyl group forms hydrogen bonds with Asn436 and Lys188 (Figure 4, top). These findings are in accord with biochemical results showing that all steps in the OADHc mechanism up to E1a-ThDP-enamine formation are unaffected by OA elongation by one carbon. The difference of approximately 8-fold in the values of *K_m_* between OA and OP binding could be explained by differences in the strengths of their interactions. For example, some hydrogen bonds are more stable than others, which could only be detected by additional computational studies once the OA binding mode is clarified by X-ray crystallography. Cryo-EM structures of the human E2o catalytic domain have been reported recently by two groups and reveal that the E2o core is composed of 24 chains organized as homotrimers into a cubic assembly [28,29]. We next modeled the *S*8-adipoyldihydrolipoamide into the E2o active site. According to reported structural studies, the E2 active sites are highly conserved in 2-oxo acid dehydrogenase complexes and are located at the interface between two 3-fold-related subunits [30]. The E2o active site resembles that of the *A. vinelandii* E2p [30] and has two substrate channels which are perpendicular to each other, one for binding the *S*8-succinyldihydrolipoamide group, and another for CoA binding. The *S*8-adipoyldihydrolipoamide approaches the E2o active site through the channel that points toward the outer surface of the E2o core, while CoA approaches the active center through a channel that is buried into the E2o core structure (Figure 4A, bottom). The highly conserved catalytic residues His357, Asp361, and Ser305 all reside in a solvent-accessible channel. On binding of the *S*8-adipoyldihydrolipoamide substrate in the active site, the *S*8 atom with a covalently attached adipoyl group is buried into the channel while the *S*6 atom, with the rest of the lipoamide substrate, remains solvent accessible (Figure 4B, bottom). The catalytic residues His357, Asp361 and Ser305′ are positioned within hydrogen-bond distance from *S*8-adipoyldihydrolipoamide. In general, the binding of the *S*8-adipoyldihydrolipoamide in this model is not different from that reported for *S*8-succinyldihydrolipoamide [29] and both resemble the structural features of the *A. vinelandii* E2p core in complex with dihydrolipoamide (LipSH_2_) and of the *A. vinelandii* E2p core in complex with CoA and (LipSH_2_) [30]. Next, the *S*8-adipoyldihydrolipoamide and CoA were modeled simultaneously in the E2o active center, showing that CoA only partially occupies the binding channel here as well (Figure 4C, bottom). In this model, the interaction network which includes the catalytic residues His357, Asp361, Ser305′, *S*-8-adipoyldihydrolipoamide and CoA in Figure 4D (bottom) is not different from that reported for *S*-8-succinyldihydrolipoamide and CoA substrates reported earlier [29], indicating that events in the E2o active center are not affected by elongation of the OG chain by two carbons in OP. While this conclusion is built on modeling studies, future structural studies of the assembled E1a-E2o, E1a-E3, and E2o-E3 sub-complexes and of an entire OADHc could provide more detailed information related to substrate channeling in the OADH.

### 2.5. The E1a-Specific Activity and the E1a Side-Reactions with OP Are Affected by E1a Substitutions Associated with AMOXAD and EoE and Showed a Great Similarity to the Behavior with OA

#### 2.5.1. Functional Characterization of the E1a Variants with Disease-Causing Mutations

The next-generation sequencing approach applied to the human genome has continued to identify disease-causing mutations in the *DHTKD1* gene [14], however, the molecular mechanisms of how the *DHTKD1* mutations lead to impaired E1a function are poorly understood. According to the X-ray structure of the human E1a reported by two research groups, the mutations identified in diseases are scattered throughout the entire E1a and may not contribute to the stabilization of the ThDP cofactor in the E1a active centers [16,28]. An initial attempt was undertaken to explain the effect of mutations on E1a function through an analysis of the stability of the recombinant E1a variants. It was concluded that most of the missense E1a variants so far identified are soluble except for Leu234Gly E1a and Ser777Pro E1a, which showed a lower yield [16,28]. Analysis of the thermal stability revealed that most of the E1a missense substitutions led to small changes in T_m_ compared to WT E1a (Gln305His, Arg455Gln, Arg715Cys, and Gly729Arg substituted E1a), indicating that their effect is not structural, while some substitutions, such as Arg163Gln, Leu234Gly, Val360Ala and Pro773Leu possess partially reduced values of T_m_ [16]. Attempts to explain impaired function of the Arg715 Cys E1a variant by using a structural approach is complicated [28] as the E1a structure by itself could not provide evidence for its impaired interaction with E2o. A multipronged systematic analysis of the pathogenic mechanism associated with mutations is needed, as was reported recently for the Gly729Arg E1a variant [26] and earlier for the human OGDHc [32]. Importantly, recent genetic studies have described a heterozygous missense mutation (c.2185G→A, p. Gly729Arg E1a) not only in cases with AMOXAD [8], but also in cases with autosomal recessive infantile-onset spinal muscular atrophy (SMA) [13], as well as in amyotrophic lateral sclerosis patients (ALS) [14].

#### 2.5.2. Studies on E1a Variants Encoded by DHTKD1 with Genetic Mutations Linked to AMOXAD and EoE

Below we present findings for the E1a variants by applying the multipronged approach used on the WT E1a above. Considering the low efficiency of E1a with OP in the overall assay of NADH production, the E1a-specific activity with OA and OP and the activities in the E1a-side reactions were analyzed first (Table 2). Among the E1a variants studied, the Gln305His E1a (66% with OA and 64% with OP) and Ser862Ile E1a (100%) behave more like WT E1a while Leu234Gly E1a (4.4% and 5.8%), Arg715Cys E1a (0.5% and 0.45%) and Arg163Gln E1a (6.9% and 4.3%) revealed low activities in the E1a-specific assay, in the E1a side-reaction of H_2_O_2_ production, and in the carboligation reaction with glyoxylate as substrate acceptor (Table 2). Interestingly, that Arg455Gln substitution, while not affecting significantly the E1a-specific activity with OA (33%) and OP (23%), reduced the H_2_O_2_ production with both OA (0.6%) and OP (3.8%) as well as the E1a activity in the carboligation reaction with OA (4.6%) and OP (3%). As is evident from Table 2, changes in the activities identified for E1a variants with OP showed a great similarity to that with OA, indicating that OP could be an efficient substrate of the E1a variants and also that these side reactions could not be ignored.

#### 2.5.3. Evidence for the Formation of a Pre-Decarboxylation Intermediate from OP by E1a Variants with Mutations

To determine which step in the E1a mechanism is affected by substitutions in Table 2, binding of the adipoylphosphonic acid (AP) in the active centers of the E1a variants with mutations was analyzed, which could inform about the formation of the 1′,4′-iminophosphonoadipoyl-ThDP, an analog of the LThDP pre-decarboxylation intermediate in Figure 1. Formation of 1′,4′-iminophosphonoadipoyl-ThDP could be monitored via a positive CD band developed at ~305–309 nm on titration by AP [16]. A positive CD band at ~305–309 nm was in evidence for Leu234Gly, Gln305His and Arg455Gln E1a’s and displayed saturation at a stoichiometric binding of AP in the active centers of the E1a variants, not different from that reported for WT E1a (Figure 5) [16]. A positive CD_305_ band was also in evidence for Pro773Leu E1a, however, no saturation could be reached on titration by AP, thus informing about weak substrate binding and/or weak stabilization of the pre-decarboxylation intermediate. 

The Arg715Cys E1a behaved differently from other E1a’s: no characteristic CD_305_ band could be detected on titration by AP in the 2–70 μM range (for comparison, the concentration of active centers of Arg715Cys E1a was ~19.4 μM). The data suggested that the Arg715Cys E1a substitution could affect the substrate binding, which is also manifested by the low activities determined in all assay systems applied (Table 2). Next, on titration of the Arg715Cys E1a by thiamin 2-thiothiazolone diphosphate (ThTTDP), an analog of ThDP, no characteristic CD band was in evidence in the 300–350 nm region of the CD spectra, suggesting that ThDP binding in the active centers of the Arg715Cys E1a variant could also be affected (Figure 6C) [33]. According to the reported structural studies of the human E1a, Arg715 is located at the twofold axis of the E1a homodimer and together with Arg712 forms a salt-bridge network with Asp677 of the opposite subunit which is conserved among 2-oxoacid dehydrogenases (Figure 6A) [28]. Nevertheless, the Arg715Cys substitution does not affect the formation of the E1a dimer according to size exclusion chromatography studies [28]. To determine the strength of interaction between the Arg175Cys E1a and E2o, the Arg175Cys E1a was titrated by the E2o ^1–173^ di-domain, comprising the LDo, the linker region and part of the E2o core domain, with an external *N*-(1-pyrene) maleimide fluorescent label attached to the sole Cys residue in this di-domain protein [26] (Figure 6D). The calculated dissociation constant of 1.13 μM was not different from *Kd* = 1.12 μM reported for WT E1a [26], suggesting that Arg175Cys E1a binding to the E2o ^1–173^ di-domain was not significantly affected. Also, the relative amounts of individual components assembled into OADHc by co-purification on immobilized metal affinity chromatography could be estimated by relative quantity calculations based on the intensity of the individual bands in PAGE. The following amounts of the components were estimated on E1a/Arg715Cys E1a assembly with E2o and E3 into OADHc: for E1a (46%), E2o (20%), E3 (20%); and for Arg715Cys E1a (7.7%), E2o (33%), E3 (32.5%), indicating that assembly of the Arg715Cys E1a with E2o and E3 components could indeed be affected (Figure 6E). Altogether, the data suggest that the Arg715Cys substitution may induce some local conformational changes in the C-terminal region of E1a that adversely affect assembly into a “correctly functioning” E1a dimer with consequences for substrate and ThDP binding, as well as for its assembly with the E2o and E3 components into OADHc. Earlier, we reported data for the Gly729Arg E1a variant and demonstrated that local conformational changes in the C-terminal region affect the E1a-E2o interaction and intermediate channeling, while not affecting the E1a function in the E1a-specific reaction and in the E1a-side reactions [26].

## 3. Materials and Methods

### 3.1. Protein Expression and Purification

Construction of plasmid encoding C-terminally His_6_-tagged E1a, expression and purification of E1a, E2o, and E3 components was as reported earlier [2]. The E1a variants with Leu234Gly, Gln305His, Arg455Gln and Arg715Cys substitutions were constructed using QuikChange^®^ II XL site-directed mutagenesis kit. The pET-22b (+) vector encoding the WT E1a as a template and the amplification primers containing the desired mutations and their complements were used for the mutagenesis reactions. The following primers and their complements were synthesized by Integrated DNA Technologies, Inc., (IDT, Coralville, IA, USA).
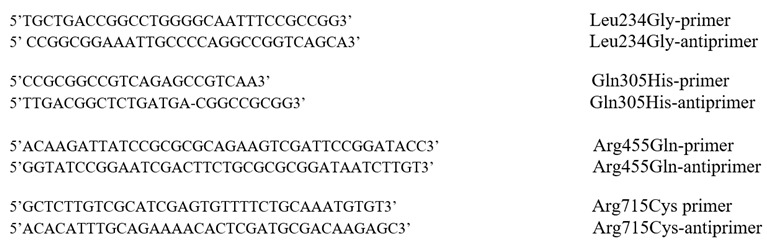


The DNA substitutions were confirmed by sequencing at GENEWIZ, LLC (South Plainfield, NJ, USA). The E1a variants with Arg163Gln, Val360Ala, and Ser862Ile substitutions were constructed using the pET28a (+) vector encoding WT E1a as a template and specific oligonucleotide primers using the QuikChange^®^ II site-directed mutagenesis kit as reported earlier [26]. The E1a and E1a variants were purified to homogeneity by using Ni-Sepharose high-performance affinity media (see SI Appendix A for E1a).

### 3.2. Overall Activity of NADH Production upon Assembly of E1a with E2o and E3

Overall activity of NADH production was measured as reported earlier [2,3,35]. For OADH complex assembly, the E1a (0.3 mg, 9.7 μM subunit concentration) in 0.1 M Tris·HCl (pH 7.5), containing 0.3 M NH_4_Cl, 0.5 mM ThDP and 2.0 mM MgCl_2_ in 0.3 mL, was mixed with E2o (0.60 mg, 46 μM subunit concentration) and E3 (1.5 mg, 100 μM subunit concentration) components at a mass ratio (μg:μg:μg) of 1:2:0.5 for 40 min at 25 °C. A 0.01–0.02 mL aliquot of the reaction mixture containing 0.005–0.01 mg of E1a and the corresponding amounts of E2o and E3 was withdrawn to start the reaction. The reaction medium contained the following in 1.0 mL: 0.10 M Tris· HCl (pH 7.5), 0.5 mM ThDP, 2.0 mM MgCl_2_, 2.0 mM DTT, 2.5 mM NAD^+^. The reaction was initiated by the addition of OA/OP (1.0 mM) and CoA (0.30 mM) after 1 min of equilibration at 37 °C. Steady-state velocities were taken from the linear portion of the progress curve. One unit of activity is defined as the amount of NADH produced (μmol∙min^−1^·mg E1a^−1^).

### 3.3. E1a-Specific Activity

The E1a-specific activity was measured in the reaction medium with the external oxidizing reagent 2,6-dichlorophenolindophenol (DCPIP) by monitoring its reduction at 600 nm, similarly to that reported earlier for E1o [3,36]. The reaction medium contained the following in 1 mL: 50 mM KH_2_PO_4_ (pH 7.0), 0.15 M KCl, 0.5 mM ThDP, 2.0 mM MgCl_2_, 1.0 mM OA/OP, 0.077 mM DCPIP (OD_600_ = 1.2), and 1% glycerol. The reaction was initiated by the addition of E1a (0.01–0.015 mg) and was recorded for 1 min at 37 °C. Steady-state velocities were taken from the linear portion of the progress curve. One unit of activity is defined as the amount of DCPIP reduced (μmol min^−1^·mg E1a^−1^).

### 3.4. Measurement for Formation of Adipoyl Semialdehyde by E1a from OP

The formation of adipoyl semialdehyde by E1a was measured in an alcohol dehydrogenase-coupled reaction assay containing: 0.1 M Tris-HCl (pH 7.5), 150 mM NaCl, 0.5 mM ThDP, 2.0 mM MgCl_2_, 0.25 mM NADH, 1 mM DTT, 1% glycerol, 0.08 mg/mL horse liver alcohol dehydrogenase (ADH), and 1 mM OP. The reaction was initiated by the addition of E1a (0.1 mg/mL) after 1 min of equilibration at 37 °C. The rate of NADH oxidation was measured at 340 nm from the initial slope of the recorded progress curve.

### 3.5. Measurement of the Rate of H_2_O_2_ Production by Fluorescence Spectroscopy

The rate of H_2_O_2_ production by E1a from OA and OP was measured using a fluorescent Amplex^TM^ UltraRed reagent in an assay reported in the literature [37] and reported by us earlier [2,26]. The reaction medium contained the following in 2.5 mL: 5 mM HEPES and 5 mM KH_2_PO_4_ (pH 7.0), 0.15 M KCl, 2.5 mM MgCl_2_, 0.25 mM ThDP, 5 units/mL of horseradish peroxidase (HRP), 25 units/mL of superoxide dismutase (SOD), 50 μM Amplex UltraRed, 1 mM OA/OP, and 0.10 mg of E1a at 37 °C. Time-dependent H_2_O_2_ production was measured on a Varian Cary Eclipse fluorescence spectrophotometer in the kinetic mode with an excitation wavelength of 560 nm and an emission wavelength of 590 nm. The reaction was initiated by the addition of E1a after 2–3 min of recording of background due to the spontaneous oxidation of Amplex UltraRed. The amount of H_2_O_2_ produced was calculated from a calibration curve that was generated with known concentrations of H_2_O_2_ using a reaction assay similar to that used in the experiment. The slope due to background oxidation of Amplex UltraRed was subtracted from all experimental curves. The activity is defined as the amount of H_2_O_2_ produced, taking into account that Amplex UltraRed reacts with HRP-H_2_O_2_ at a 1:1 stoichiometry (nmol min^−1^ mg E1a^−1^).

### 3.6. Circular Dichroism Spectroscopy for Detection of Carboligase Products, (S)-2-Hydroxy-3-Oxo-Octanedioic and (S)-8-Ethoxy-7-Hydroxy-6,8-Dioxo-Octanoic Acid

The CD spectra of the (*S*)-2-hydroxy-3-oxo-octanedioic and (*S*)-8-ethoxy-7-hydroxy-6,8-dioxo-octanoic acid products formed by E1a from OA or OP as substrate donors and glyoxylate or ethyl glyoxylate as substrate acceptors were recorded on a Chirascan CD spectrometer (Applied Photophysics, Leatherhead, UK) in a 1-cm path length cell in the near-UV (290–450 nm) wavelength region as reported by us earlier [2,35]. Briefly, the E1a (2.0 mg/mL, 19.4 μM active centers) in 2.4 mL of 5 mM HEPES and 5 mM KH_2_PO_4_ (pH 7.0) containing 0.15 M NaCl, 0.2 mM ThDP, 2.5 mM MgCl_2_ and 5% glycerol was incubated with 500 μM OP and 10 mM glyoxylate or 10 mM ethyl glyoxylate as substrate acceptor for 15 h at 4 °C. Protein was separated from the reaction mixture by using an Amicon^®^ Ultra—4 centrifugal filter unit (Millipore) and the CD spectra of the reaction mixture free of the protein were recorded in the 290–450 nm wavelength region. The progress curves of (*S*)-2-hydroxy-3-oxo-octane-dioic acid formation were recorded by CD at 280 nm in the kinetic mode. The reaction was initiated by the addition of E1a (0.1 mg/mL) and was measured for 500 s at 37 °C. The amount of (*S*)-2-hydroxy-3-oxo-octanedioic acid formed was calculated by using the molar ellipticity [Θ] of 3450 deg·cm^2^·dmol^−1^ reported for (*R*)-acetoin at 278 nm [38]. Steady-state velocities were calculated from the linear region of the progress curves and were plotted versus different concentrations of OP.

### 3.7. Circular Dichroism Titration of E1a and E1a Variants by Adipoylphosphonic Acid and by Thiamin 2-Thiothiazolone Diphosphate 

In titration experiments with AP, the E1a or E1a variants (2.0 mg/mL, concentration of active centers of 19.4 μM) in 100 mM HEPES (pH 7.5) containing 0.15 M NaCl, 0.5 mM ThDP, 2.0 mM MgCl_2_ and 6% glycerol at 25 °C were titrated by AP (2–70 μM) and CD spectra were recorded in the near-UV region at 290–450 nm. The intensity of the CD band at 305 nm was plotted versus the concentration of AP and data points were fitted to a modified Hill equation [CD_305_ = yo + ((CD_305max_ × x^n^)/(S_0_._5_^n^ + x^n^))].

Titration experiments with ThTTDP were as reported previously [39].

### 3.8. Reductive Acylation of LDo by E1a and OA or OP In Vitro as Detected by Fourier Transform Mass Spectrometry

Reductive glutarylation (or adipoylation) of the E2o lipoyl domain (LDo) was as reported by us earlier [2,27,40,41]. In a steady-state experiment, the LDo (100–150 μM) was incubated with E1o (0.20 μM) in 50 mM NH_4_HCO_3_ (pH 7.5) containing 0.5 mM MgCl_2_ and 0.10 mM ThDP in 0.35 mL. The reaction was initiated by the addition of 2.0 mM OA or 5 mM OP. An aliquot of 10 μL of the reaction mixture was withdrawn at different times of incubation and was quenched into 1 mL of 50% methanol, 0.1% formic acid. Samples were analyzed for relative amounts of the glutarylated (or adipoylated) and unglutarylated (or unadipoylated) forms of LDo by Fourier transform mass spectrometry (FT-MS). The ratio of intensity of the glutarylated (or adipoylated) LDo versus that of the total intensity (sum of glutarylated and unglutarylated) (or sum of adipoylated plus unadipoylated) was plotted against time. The rates of the reaction were calculated from the initial slope.

### 3.9. Enzymatic Synthesis of Adipoyl-CoA

For the enzymatic synthesis of adipoyl-CoA, E1a (0.15 mg) was assembled with E2o (0.30 mg) and E3 (0.75 mg) into OADHc in 0.15 mL of 0.1M Tris·HCl (pH 7.5) containing 0.3 M NH_4_Cl, 0.5 mM ThDP and 2.0 mM MgCl_2_ at 25 °C [26]. After 40 min of incubation, an aliquot containing 0.02 mg of E1a and the corresponding amounts of E2o and E3 was withdrawn and placed into 0.4 mL of the reaction assay containing 0.1 M Tris·HCl (pH 7.5) and all components necessary for the overall NADH assay: 2.0 mM MgCl_2_, 0.5 mM ThDP, 2.0 mM DTT, and 2.5 mM NAD^+^. The reaction was initiated by the addition of OP (2.0 mM) and CoA (200 μM) after 1 min of equilibration at 37 °C in the Eppendorf ThermoMixer.

The reaction was terminated at different times by acidification on the addition of 2.5% TFA in H_2_O to a final concentration of 0.1% TFA, and samples were kept at 4 °C. The adipoyl-CoA formed in the reaction was purified according to the protocol reported in the literature [42]. The samples were analyzed in a negative mode using a 7T Bruker Daltonics FT-MS instrument and the following acquisition parameters: negative mode, 30 averaged scans, ion accumulation time of 0.5 s, source ESI, number of laser shots 100, laser power 30.0%. The spectra were analyzed by using Bruker Compass DataAnalysis 4.3 software (Bruker Daltonics, Bremen, Germany) and were base line subtracted and smoothed. Data points were analyzed by using SigmaPlot Version 10.0 software by Systat Software Inc., and equations are presented under Fig. Legends.

### 3.10. Fluorescence Titration of Pyrene-Labeled E2o^1–173^ Di-Domain by Arg715Cys E1a

The fluorescence spectra were recorded at 25 °C by using a Varian Cary Eclipse fluorescence spectrophotometer in the reaction medium containing 100 mM HEPES (pH 7.5), containing 0.15 M NaCl and 1% glycerol. The E2o^1–173^ di-domain (50 μM) was labeled with N-(1-pyrene) maleimide according to aur reported protocol [26,32]. The titration experiment was conducted as reported elsewhere [26,32].

### 3.11. Ligand Docking

The crystal structure of E1a was prepared for docking using the Protein Preparation Wizard in Maestro (Schrödinger) as recently reported [31]. Induced fit docking was carried out with Glide/Prime (Schrödinger) using the standard protocol and docking box centered on the active site of the crystal structure. Figures were generated using PyMOL Molecular Graphics System version 2.3.2. (Schrödinger, LLC, New York, NY, USA) [31]. *S*8-adipoyl-dihydrolipoamide and CoA docking into E2o active center was as recently reported by Nagy et al. [29]. Coordinates for dihydrolipoamide and CoA were obtained from the X-ray structures of *A. vinelandii* E2p (PDB IDs: 1EAE and 1EAD) [30]. After placing and optimally orienting the dihydrolipoamide and CoA in the respective channels, one round of energy minimization was carried out using Chimera [43] and the parameters recently reported by Nagy et al. [29]. Another two rounds of energy minimization were performed after manually attaching the adipoyl group to the *S*8 atom of the dihydrolipoamide by using the same parameters. To validate the substrates’ docking into the corresponding channels of E2o, a similar structure of the *A. vinelandii* E2p with dihydrolipoamide and CoA both present was generated in accordance with crystallographic studies [29,30]. The model structures for *A. vinelandii* E2p and E2o, after energy minimization imposed on substrates only, displayed similar channel structures and substrate binding modes. Molecular visualization was performed by PyMol [31]. Solvent accessible channels were analyzed using the program Caver 3.0 [44].

## 4. Conclusions

The findings in this paper add to our understanding of the OADHc function and of the molecular mechanisms leading to the pathogenesis associated with *DHTKD1* encoded variants. To understand the molecular mechanisms of how the identified rare DHTKD1 variants affect the E1a function, a few approaches have been reported in the literature: (i) structural mapping of DHTKD1 variants encoded by disease-causing mutations to understand their role on E1a function [6,16,28]; (ii) reconstruction recombinantly of DHTKD1 (E1a) variants encoded by missense *DHTKD1* mutations followed by analysis of the protein solubility and thermostability, as well as assessment of their E1a-specific activity in vitro to probe their pathogenicity [16]. Jordan’s group reported a multipronged systematic analysis of the mechanistic consequences of the c.2185G > A *DHTKD1* encoded (p. Gly729Arg E1a) variant, where the detailed analysis of the reaction rates in the OADHc pathway in vitro was supplemented with results on the E1a-E2o interaction from chemical cross-linking and hydrogen-deuterium exchange MS [26], leading to an important conclusion that correct positioning of the C-terminal E1a region is essential for intermediate channeling in the E1a-E2o binary subcomplex [26]. To provide a more complete understanding of intermediate channeling in the OADHc, future efforts should be focused on structural studies of the OADHc by Cryo-EM or X-ray, which have not been successful so far. Herein, we presented a multipronged analysis of the E1a variants encoded by *DHTKD1* with genetic mutations linked to AMOXAD and EoE, and suggested a molecular mechanism for the Arg715Cys E1a variant which potentially could explain its pathogenesis. The important conclusions from our studies in this paper combined with the results reported by us earlier [2,3,16,35,45] are the following:The functional promiscuity of the human E1a component has been demonstrated and adds to our understanding of the OADHc function in the L-lysine catabolism pathway. According to biochemical and structural findings, the E1a active centers accept OP with a one-carbon longer chain compared to the natural substrate OA, with no significant effect on the rate of decarboxylation and on the formation of the E1a-ThDP-enamine intermediate, as well as on all three side-reactions derived from the E1a-enamine intermediate. Importantly, the findings indicate that E1a is a better source of H_2_O_2_ production from OA and OP than E1o with its natural substrate OG, a finding that may potentially affect mitochondrial structure and function.The E2o component, an important enzyme in the TCA cycle, is known as a gatekeeper in intermediate channeling in 2-oxo acid dehydrogenase complexes [29,30,46,47,48,49,50]. Our findings suggest involvement of the E2o in the rate-limiting step at two levels in the mechanism, specifically, by controlling the rate of transfer of the acyl group between the E1a and E2o components (so called reductive acylation) as generally accepted, and by controlling the rate of transfer of the acyl group from *S*8-acyldihydrolipoyl-E2o to CoA in its active centers. At the same time, the rate of recycling of dihydrolipoyl-E2o to lipoyl-E2o proceeds faster [26,40,41].Evidence is presented that the E2o component of the 2-oxoglutarate dehydrogenase complex in the TCA cycle can produce acyl-CoA thioesters of different lengths, such as succinyl-CoA (C4), glutaryl-CoA (C5) and, less efficiently, adipoyl-CoA (C6), indicating its promiscuity, not recognized before.The E1a variants with mutations associated with AMOXAD and EoE affected the E1a-specific and the E1a-enamine side reactions with OP and showed a great similarity to that observed with OA. The mechanism underlying diminished E1a activity due to the Arg715Cys substitution is suggested. The data point to local conformational changes in the C-terminal region of the E1a homodimer, with consequences for ThDP cofactor and substrate binding, as well as for its assembly with E2o and E3.

## Data Availability

Not applicable.

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
