# Peer review of "Functional Versatility of the Human 2-Oxoadipate Dehydrogenase in the L-Lysine Degradation Pathway toward Its Non-Cognate Substrate 2-Oxopimelic Acid"

_ijms, 2022, doi:10.3390/ijms23158213_

Round 1
Reviewer 1 Report
This manuscript by Nemeria et. al. found the conclusions are identification of (i) Functional promiscuity of E1a and (ii) The ability of the E2o to form acyl-CoA products derived from homologous 2-oxo acids with five, six, and even seven carbon atoms. The manuscript is well organized, and the conclusions are supported by the analysis and discussions. However, there are several points in this manuscript that are not clear enough and are required to be modified before acceptance.
1. Introduction: Are there other methods that should be discussed in the introduction as well, how does this approach compare to those?
2. Please improve the resolution of figures 2&6.
3. The conclusion looks fine, and the main limitation also should be discussed as well.
4. There are some grammatical errors in this manuscript such as continuously forgetting to add ‘a’ or ‘the’ before a specific word which limits the clarity of the author’s writing. Check the language issues.
Reviewer 2 Report
In this manuscript, the authors cloned and expressed the enzymes in the L-lysine catabolism and investigated the possible mechanism of the enzyme complex. This manuscript will benefit the understanding of the enzyme complex functions and the related disease. The manuscript is well organized and could be accepted after appropriate modification.
1) The specific activity and kinetic parameters of the enzymes are highly dependent on the protein purity. If possible, the presence of SDS-PAGE images of the mutants and enzymes would be good.
2) The version of PyMOL could be indicated.
